# Survival Improvements in Advanced Hepatocellular Carcinoma with Sequential Therapy by Era

**DOI:** 10.3390/cancers15215298

**Published:** 2023-11-06

**Authors:** Yoshiko Nakamura, Masashi Hirooka, Atsushi Hiraoka, Yohei Koizumi, Ryo Yano, Makoto Morita, Yuki Okazaki, Yusuke Imai, Hideko Ohama, Kana Hirooka, Takao Watanabe, Fujimasa Tada, Osamu Yoshida, Yoshio Tokumoto, Masanori Abe, Yoichi Hiasa

**Affiliations:** 1Department of Gastroenterology and Metabology, Ehime University Graduate School of Medicine, Toon 791-0295, Japan; yoshikon@m.ehime-u.ac.jp (Y.N.); hiasa@m.ehime-u.ac.jp (Y.H.); 2Gastroenterology Center, Ehime Prefectural Central Hospital, Matsuyama 790-0024, Japan; hirage@m.ehime-u.ac.jp (A.H.);; 3Department of Gastroenterology, Takarazuka City Hospital, Takarazuka 665-0827, Japan; 4Department of Gastroenterology and Metabology, National Hospital Organization Ehime Medical Center, Toon 791-0281, Japan

**Keywords:** hepatocellular carcinoma, treatment outcome, molecular targeted therapy, immune checkpoint inhibitors

## Abstract

**Simple Summary:**

Systemic chemotherapies have revolutionized therapeutic paradigms for hepatocellular carcinoma (HCC) over the past decade. Various regimens have progressively become standard-of-care in clinical practice. However, there are no reports regarding the impact of multiple sequential therapies including immune-oncologic agents on their outcomes. This study investigated the change, over three time periods, in survival for patients with unresectable HCC, and the effects of sequential treatments on their outcomes. The results showed that the number of patients who received more than two lines steadily increased, and the overall survival significantly improved over time. Surprisingly, the 3-year survival rate increased from 12.1% in the early period to 44.4% for the most recent period. Using less than three lines, the non-objective response of the first line and extrahepatic metastasis were identified as the strongest drivers of a worse prognosis. Sequential treatment post-progression is valuable for prolonging survival.

**Abstract:**

Treatment modalities for advanced hepatocellular carcinoma (HCC) have changed dramatically, with systemic therapy as the primary option. However, the effect of sequential treatment on prognosis remains unclear. This retrospective study included patients who began systemic therapy between 2009 and 2022. The patients were separated into three groups according to systemic therapy commencement. The number of therapy lines, treatment efficacy, and overall survival (OS) were compared. Multivariate analyses of the prognostic factors were analyzed using the Cox proportional hazards model. Overall, 336 patients were included (period 1: 2009–2013, *n* = 86; period 2: 2014–2018, *n* = 132; period 3: 2019–2022, *n* = 118). A significant etiological trend was observed with decreasing viral hepatitis-related HCC and increasing non-viral hepatitis-related HCC. Across periods 1–3, the proportion of patients who were administered >2 lines progressively increased (1.2%, 12.9%, and 17.0%, respectively; *p* < 0.001) and the median OS was significantly prolonged (14.3, 16.8, and 31.0 months; *p* < 0.001). The use of <3 lines, the non-complete and partial response of the first line, modified albumin–bilirubin at grade 2b or 3, an intrahepatic tumor number ≥ 5, extrahepatic metastasis, and alpha-fetoprotein at ≥400 ng/mL were the strongest factors associated with shorter OS. Sequential therapies have contributed to significant improvements in HCC prognosis, suggesting that sequential treatment post-progression is worthwhile for better survival.

## 1. Introduction

Hepatocellular carcinoma (HCC) remains the sixth most common cancer and is a leading cause of liver-related mortality [1,2]. The incidence rate of HCC is still growing globally, with an estimate of over 1 million cases diagnosed by 2025 [3,4]. HCC has been reported as the fastest growing pathogenesis of cancer-related deaths in the US since the early 2000s [5,6]. However, the number of incidences and deaths has been decreasing in Japan due to the establishment of screening tests, treatments, and transmission prevention measures for hepatitis B virus (HBV) and hepatitis C virus (HCV) [2]. However, as in the US, non-alcoholic steatohepatitis (NASH) has become the growing etiology of HCC owing to the increasing prevalence of obesity [7].

Recent advances in chemotherapy have markedly improved treatment outcomes for malignancies. Molecular targeted agents (MTAs) and immune checkpoint inhibitors have revolutionized therapeutic paradigms for various tumor types. Some reports have documented that the increased utilization of systemic therapy options has provided patients with a better chance of survival among various types of malignancies [8,9].

Currently, two-thirds of patients with HCC are possible candidates for curative treatments, while one-third of patients receive non-curative treatments [10]. According to the Japan Society of Hepatology (JSH), since the first approval of sorafenib in May 2009, six systemic treatment regimens for HCC have been subsequently approved in Japan [11]. Additionally, a further novel combination therapy (involving durvalumab plus tremelimumab) has been recently approved as a first-line therapy [12].

Systemic therapy has been globally recommended for advanced-stage HCC. Additionally, it has been recently introduced to intermediate-stage HCC to achieve curative conversion therapy, instead of transcatheter arterial chemoembolization (TACE) [13]. Furthermore, the American Association for the Study of Liver Disease (AASLD) consensus statement, updated in 2020, recommends upfront systemic therapy in addition to TACE [14]. 

These changes may have contributed to the prolonged subclinical survival rates observed over the last decade. However, few studies have addressed the temporal trends of patients with HCC receiving systemic therapy in real-world settings. Kobayashi et al. previously addressed patients with advanced-stage HCC who had multiple MTAs had prolonged prognoses [15]. In contrast, there was no significant difference among patients with non-advanced-stage HCC. However, in their report, the outcome of patients treated with the present first-line therapy, atezolizumab and bevacizumab (Atezo + Bev), was not included. Furthermore, the effect of systemic therapy for patients with non-advanced stage HCC had not been assessed. Therefore, in this study, we aimed to evaluate trends in advanced HCC presentation and treatment between 2008 and 2022, and investigate the impact of sequential treatments on patient outcomes. 

## 2. Materials and Methods

### 2.1. Patients and Systemic Treatment

A total of 378 patients began systemic therapy for HCC between May 2009 and December 2022 at Ehime University Hospital and Ehime Prefectural Central Hospital. According to the diagnostic criteria of the AASLD, HCC was diagnosed based on radiological findings through the assessment of dynamic contrast-enhanced computed tomography (CT) and magnetic resonance imaging (MRI) scans and/or histological findings. The Barcelona Clinic Liver Cancer (BCLC) staging system was used to determine the tumor progression status at the start time of systemic therapy. The approval dates of sorafenib, regorafenib, lenvatinib, ramucirumab, Atezo + Bev, and cabozantinib by the Japanese regulatory authority, the Pharmaceuticals and Medical Devices Agency, were May 2009, June 2017, March 2018, June 2019, September 2020, and November 2020, respectively. 

The inclusion criteria were (1) BCLC-B and BCLC-C patients, (2) systemic treatment starting between May 2009 and December 2022, and (3) the use of the following systemic agents: sorafenib, lenvatinib, Atezo + Bev, regorafenib, cabozantinib, and ramucirumab. The exclusion criteria were (1) no liver imaging with contrast media, (2) an observation period of less than 1 month, and (3) a lack of data for evaluation.

According to the manufacturer’s instructions, the initial doses of the orally administered agents were as follows: (1) sorafenib: 800 mg/day, (2) lenvatinib: 8 mg/day to patients with body weight (BW) < 60 kg or 12 mg/day to patients with BW ≥ 60 kg, (3) regorafenib: 160 mg/day once a day for 3 weeks followed by a 1-week break, and (4) cabozantinib: 60 mg/day. However, some cases were initiated at lower doses at the physician’s discretion. Intravenous agents were delivered as follows: (1) Atezo (1200 mg) + Bev (15 mg/kg) administered intravenously every 3 weeks, and (2) ramucirumab: 8 mg/kg intravenously injected every 2 weeks to patients with a serum AFP level of ≥400 ng/mL. 

If any unacceptable adverse events (AEs) were observed, dose reductions or discontinuations were carried out until the symptoms resolved to grade 2, according to the National Cancer Institute Common Terminology Criteria for Adverse Events, version 4.0. Subsequent therapy was selected according to the physician’s consideration, when treatment was ceased due to progressive disease (PD) or severe AEs. 

### 2.2. Study Design

In total, this retrospective study included 336 patients. According to the start date of the first systemic therapy, the patients were divided into the following three groups: period 1, between 2009 and 2013, utility of sorafenib alone; period 2, between 2014 and 2018, availability of regorafenib and lenvatinib in addition to sorafenib; and period 3, between 2019 and 2022, additional approval of ramucirumab, Atezo + Bev, and cabozantinib (Figure 1). 

This study complied with the ethical principles outlined in the Declaration of Helsinki. Prior to treatment initiation, written informed consent was obtained from each patient. 

### 2.3. Evaluation of Therapeutic Efficacy, Hepatic Function, and Etiology

The Modified RECIST (mRECIST) criteria were used to evaluate the therapeutic response (complete response (CR), partial response (PR), stable disease (SD), and progressive disease (PD)). The initial assessment of therapeutic response was performed using dynamic enhanced CT or MRI approximately 4–6 weeks after initiation. Thereafter, it was performed at every 8–12-week interval, although in some cases, it was performed at longer intervals, at the physicians’ discretion. The disease control rate (DCR) and objective response rate (ORR) were defined as the sum of the CR, PR, and SD rates, and the CR and PR rates, respectively. The time to progression (TTP) was defined as the duration between the date of the beginning of the first systemic treatment and the date of PD onset. The overall survival (OS) was defined as the time from the beginning of the first systemic treatment to the date of death or last follow-up. 

The albumin–bilirubin (ALBI) score, modified ALBI (mALBI) grades, and Child–Turcotte–Pugh classification were calculated to assess hepatic reserves. Viral liver disease was determined in cases where hepatitis B surface antigens and/or hepatitis C virus antibodies were detected. Non-viral liver disease was defined as a negative result for these tests.

### 2.4. Statistical Analysis

Data are represented as median and range for continuous variables and as percentages for categorical variables. Differences were analyzed using the Kruskal–Wallis test for continuous variables and the chi-square test or Fisher’s exact test for qualitative variables. The PFS and OS were estimated using the Kaplan–Meier method, and between-group comparisons of PFS was performed using the log-rank test. Between-group comparisons of OS were also performed using both the log-rank test and the Cox proportional hazards model. To evaluate the impact of various factors on OS, univariate and multivariate analyses were performed using the Cox proportional hazards model. 

Statistical significance was defined as *p* < 0.05. Missing data were excluded from analysis. Statistical analyses were performed using the STATA/BE version 17.0 software (Stata Corp LLC, College Station, TX, USA) and JMP version 11.2.0 software (SAS institute, Cary, NC, USA).

## 3. Results

### 3.1. Patient Characteristics

We enrolled 336 patients who underwent systemic therapy for HCC. The patient characteristics are shown in Table 1. The median age was 71 years (range 19–89), and 282 of the patients (83.9%) were men. The burden of viral liver disease was observed in 237 patients (71.0%). The number of patients with mALBI grades 1 or 2a at baseline was 234 (69.6%). The median maximum intrahepatic tumor size was 28 mm, and approximately half of the patients had five or more intrahepatic tumors. Macrovascular invasion and extrahepatic spread were observed in 85 (25.3%) and 157 patients (46.7%), respectively. The number of patients with BCLC-B and BCLC-C was 125 (37.2%) and 211 (62.8%), respectively. The median AFP level was 54 ng/mL. The number of systemic therapy lines was one in 248 patients (73.8%), two in 50 patients (14.9%), and more than three in 38 patients (11.3%). 

### 3.2. Patient Characteristics within Each Group

The number of patients in periods 1, 2, and 3 was 86, 132, and 118, respectively (Figure 1, Table 2). The three groups were similar in terms of median age (67 vs. 71 vs. 71 years; period 1 vs. period 2 vs. period 3, respectively), sex (male: 89.5 % vs. 83.3% vs. 80.5%), PS (0: 86.0% vs. 81.8% vs. 82.2%), ALBI grade (2b or 3: 29.1% vs. 32.6% vs. 28.8%), and BCLC stage (C: 69.8% vs. 58.3% vs. 62.7%). However, there was a significant trend in etiology between these three groups. The percentage of viral hepatitis-related HCC was found to decline over time. Conversely, a continuous increase in non-viral-hepatitis-related HCC was also observed (non-viral: 16.3% vs. 29.5% vs. 39.0%, *p* = 0.002). In line with this etiology trend, a significant difference in body mass index between the period 1 and 3 groups was also observed (22.2 kg/m^2^ in period 1 to 24.1 kg/m^2^ in period 3; *p* = 0.005). Additionally, the ratio of the number of intrahepatic tumors ≥ 5 was found to significantly decrease (58.1% vs. 59.1% vs. 44.1%, *p* = 0.037). Moreover, an increase in levels of patients with no extrahepatic spread was observed; however, this difference was not significant (46.5% vs. 52.3% vs. 59.3%). 

### 3.3. Differences in Systemic Therapy Utilization

Most patients in period 1 (97.6%), 69.7% in period 2, and 61.1% in period 3 received only one line of therapy. However, a significantly higher proportion of use of three or more lines was observed over time (1.2% vs. 12.9% vs. 16.9%; *p* < 0.0001) (Table 2, Figure 2). 

Prior to the approval of lenvatinib use, all patients in period 1 received sorafenib as the standard of care. Only two patients in period 1 received sequential chemotherapy, due to hand–foot syndrome and disease progression. The sequential therapies used were TS-1 and *UFT* due to the absence of second-line treatment options with proven effectiveness (Figure 2). In period 2, the percentages of patients receiving lenvatinib and sorafenib as first-line therapies were 15% and 85%, respectively. As a consequence of adding subsequent agents with beneficial evidence, both the number of patients receiving sequential treatments and the variety of administered agents increased. Finally, 47% of patients in period 3 were treated with Atezo + Bev as the first-line therapy. As previously observed in these trends, the number and variety of systemic therapy lines increased.

### 3.4. Therapeutic Efficacy of Systemic Therapy

The radiotherapeutic response to systemic therapy according to mRECIST is shown in Table 3. The ORR and DCR were significantly higher in period 3 compared to those in the other two periods (ORR/DCR: 14.9/31.1 vs. 19.2/41.3 vs. 38.0/76.0, respectively; *p* = 0.001, <0.0001). 

The median TTP was 4.6 months (Figure 3a) and the median OS was 19.9 months (Figure 4a) in the entire cohort. Both the TTP and OS in period 3 were significantly higher than those in the other two periods (median survival time: 14.3 vs. 16.8 vs. 31.0 months, respectively; *p* < 0.0001) (Figure 3b and Figure 4b). Likewise, the 3-year OS increased from 12.1% for period 1 to 22.0% for period 2 and 44.4% for period 3. The TTP was comparable between periods 1 and 2; however, a significantly gradual improvement in OS was observed in period 2, suggesting that the use of sequential therapy might facilitate this benefit.

### 3.5. Overall Survival among BCLC-B Patients and BCLC-C Patients

We additionally analyzed the overall prognosis of the BCLC-B and BCLC-C subgroups. The characteristics of the BCLC-B patients are shown in Appendix A. The median OS showed a gradual improvement over time, and those in period 3 had the longest prognosis among both the BCLC-B patients (median OS: 15.4 vs. 21.4 vs. 37.6 months, respectively; *p* = 0.0001) (Figure 5a) and the entire cohort. 

Appendix A shows the characteristics of patients with BCLC-C. Period 3 had the highest OS rate among the three groups in the BCLC-C population, although this difference was not significant (median OS: 13.1 vs. 15.0 vs. 21.7 months, respectively; *p* = 0.0971) (Figure 5b).

### 3.6. Predictors of Mortality

The results of the univariate and multivariate analyses for OS are summarized in Table 4. The univariate analyses found that the mALBI grade, intrahepatic tumor number, macrovascular invasion, AFP value, the number of systemic therapy lines, and therapeutic efficacy of first-line therapy were associated with OS. In contrast, age, sex, maximum intrahepatic tumor size, and etiology were not significantly associated with OS. 

The Cox proportional hazards model was subsequently used to estimate the hazard ratios (HRs) relating the risk factors to survival outcomes. A multivariate analysis showed that mALBI grades 2b or 3 (HR = 1.823; 95% CI, 1.325–2.508; *p* < 0.001), an intrahepatic tumor number ≥ 5 (HR = 1.859; 95% CI, 1.327–2.604; *p* < 0.001), the presence of macrovascular invasion (HR = 1.504; 95% CI, 0.953–2.37; *p* = 0.007), extrahepatic metastasis (HR = 2.297; 95% CI, 1.378–3.828; *p* = 0.001), an AFP value ≥ 400 ng/mL (HR = 1.622; 95% CI, 1.216–2.164; *p* < 0.001), the use of less than three lines of therapy (HR = 2.776; 95% CI, 1.751–4.402; *p* < 0.001), and non-CR and PR of first-line therapy (HR = 2.516; 95% CI, 1.704–3.716; *p* < 0.001) were identified as independent factors for worse OS.

## 4. Discussion

This retrospective study highlights the gradual improvements in the OS of patients with HCC receiving systemic therapy over the last decade, when stratified according to the start time of systemic therapy. This study makes a novel contribution to the literature by clarifying that the most predictive factors for a better survival rate were the utilization of three or more sequential therapies and the therapeutic efficacy of the chosen first-line therapy.

Systemic therapies, including immune checkpoint inhibitors, tyrosine kinase inhibitors (TKIs), and MTAs, have been developed for use in advanced-stage HCC. Recent advances have further expanded their application in clinical practice. The SHARP trial [16] and the Asia–Pacific trial [17] demonstrated the superiority of sorafenib over placebos in terms of OS, which was a breakthrough in HCC treatment in 2007. In 2017, lenvatinib was shown to be non-inferior to sorafenib [18] and was approved as another standard first-line setting drug, whereas regorafenib [19], cabozantinib [20], and ramucirumab [21] were shown to be superior to placebos in patients receiving sorafenib and were established as sequential-line treatments. In addition, the IMbrave 150 trial demonstrated that the Atezo + Bev combination was the first regimen to improve OS compared to sorafenib [22], leading it to become the first-line therapy currently recommended for use in advanced HCC. In keeping with this trend, we observed that systemic therapy is being introduced earlier in the treatment of these patients. Notably, the number of intrahepatic tumors and the proportion of patients with extrahepatic spread in period 1 tended to be greater than in the two other periods (Table 1). Additionally, the number of patients in period 1 was less than in the others (Table 1). 

Over the duration of this study, we observed a trend where both the number of approved sequential chemotherapy lines and the proportion of patients receiving multi-line chemotherapy increased. Notably, multivariate analysis indicated that the use of three or more lines was the strongest predictive factor for OS, suggesting that post-progression sequential treatment is beneficial to prolong prognosis. This finding is consistent with the trend reported by Kobayashi et al. [15], who demonstrated a correlation between OS and the duration of TKI/MTA therapy. Some other previous studies have also reported similar findings [23,24]. Additionally, we confirmed improvements in OS only among patients whose first-line treatment was sorafenib (median OS: 14.3 months vs. 18.6 months vs. 26.3 months; period 1 vs. period 2 vs. period 3, respectively; data not shown). A multivariate analysis showed a similar result: that the utility of more than two lines (HR, 3.15; 95% CI, 1.744–5.69, *p* < 0.0001) and non-ORR of the first line (HR, 3.16; 95% CI, 1.788–5.60; *p* < 0.0001) were the strongest predictive factors, suggesting that sequential therapy is valuable, regardless of the first-line regimen. 

According to a nationwide follow-up survey of primary liver cancer in Japan by the Liver Cancer Study Group of Japan [25], the median OS of patients with Child–Pugh grade A who received systemic therapy between 2012 and 2013 was 12.6 months. The median OS of patients in 2009–2013 (period 1) in our study was 14.3 months (Figure 4b). However, some discrepancies were found among the BCLC-C patients. A nationwide report showed that the median OS of patients with naïve HCC having macrovascular invasion or metastasis was 7.72 months, whereas the median OS of patients treated between 2009 and 2013 in our study was 13.1 months (Figure 5b). As reported by Kobayashi et al., the median OS among BCLC-C patients in 2009–2012 was 8.7 months, and in 2013–2016, this was 10.7 months [15]. These differences could be due to several factors such as hepatic reserve, the degree of vascular invasion, naïve or recurrent treatment, or selection bias. The notable prognosis in period 1 could be one of the reasons why no significant differences in OS were observed among patients with BCLC-C. Moreover, 48% of BCLC-C patients in period 3 were undergoing treatment at the time of reporting this study, and the observation period was short. Therefore, further investigation of OS in period 3 is needed. 

Interestingly, over time, we observed a clear benefit of systemic therapy in patients with BCLC-B. As mentioned previously, the most drastic paradigm change is the strategy for the BCLC-B stage. Since it was initially proposed in 2011 [26], the principle of TACE refractoriness and unsuitability was carried out. Then, early switching to combination with molecular targeted therapy was shown to be effective [27]. This successful strategy in combination with TACE has been widely accepted in real-world settings, and most BCLC-B patients in this study received TACE in combination with systemic therapy (data not shown). In line with these trends, the survival of BCLC-B patients has markedly improved [14,28,29]. Therefore, it is possible that the change in adapting to TACE may influence the survival of BCLC-B patients in this study. 

Regarding therapeutic efficacy, Atezo + Bev therapy may contribute to the promising outcomes recorded in period 3 by causing tumor shrinkage, unlike other targeted therapies. Hiraoka et al. reported that patients with HCC who received Atezo + Bev as a first-line treatment may have a better prognosis than those who received lenvatinib [30]. Of note, we encountered some patients who underwent Atezo + Bev therapy followed by curative conversion therapy who achieved pathological CR. Furthermore, combination immunotherapies (durvalumab plus tremelimumab) have previously been used in clinical practice. Therefore, we can expect patients with unresectable HCC to have prolonged survival. 

One of the major findings of this study was that the etiological cause of HCC changes over time, with non-viral-induced HCC increasing dramatically, even in patients with advanced HCC. Tateishi et al. also confirmed that the proportion of non-viral etiology in patients with HCC has increased in the past two decades, due to the increasingly obese male population in relation to lifestyle changes [7]. Additionally, by using nationwide registry data, Okushin et al. demonstrated that the percentages of patients with HCC with HBV, HCV, and non-viral etiology on admission between April 2018 and January 2021 were 11.9%, 36.2%, and 42.6%, respectively [31]. These numbers were similar to those found in period 3 of our study. We believe that the increase in non-viral-induced HCC and the decrease in viral-induced HCC are likely due to lifestyle and metabolic factors, including alcohol consumption, obesity, elevated cholesterol levels, and novel anti-viral medications. The JSH guidelines published in 2020 suggest that surveillance for HCC should be demonstrated in high-risk populations, which has been found to be successful among some cohorts. Furthermore, according to Kim et al., HCC diagnosis via surveillance was strongly associated with decreased mortality. Nevertheless, diagnoses based on surveillance rates and the percentage of curative treatments were not found to change significantly [32]. Therefore, there is an urgent need to establish effective screening systems, especially those related to NASH.

This study has several limitations. First, as this was a retrospective study, there are some confounding factors based on the era background, e.g., the numbers of approval agents, physicians’ management and accessibility, and other treatment modalities, that may have affected the outcome and introduced some biases. Therefore, we were not able to compare the precise efficacy of sequential therapy by era. Second, the study was conducted at only two tertiary hospitals, which may limit its generalizability. Third, the observation time of period 3 was short. To reach a distinct conclusion, a longer observation period with a larger number of patients is required. 

## 5. Conclusions

The prognosis of patients with HCC receiving systemic therapies has improved significantly over the past decade. The use of three or more lines of therapy is one of the most predictive factors associated with better OS, and sequential treatment post-progression is valuable to prolong survival. 

## Figures and Tables

**Figure 1 cancers-15-05298-f001:**
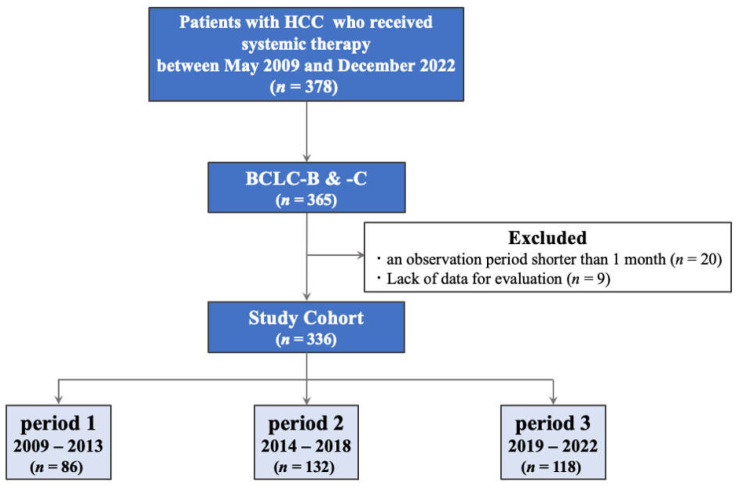
Study design flowchart showing the identification of 378 patients. Three hundred and thirty-six patients with HCC were ultimately enrolled in this study. HCC, hepatocellular carcinoma.

**Figure 2 cancers-15-05298-f002:**
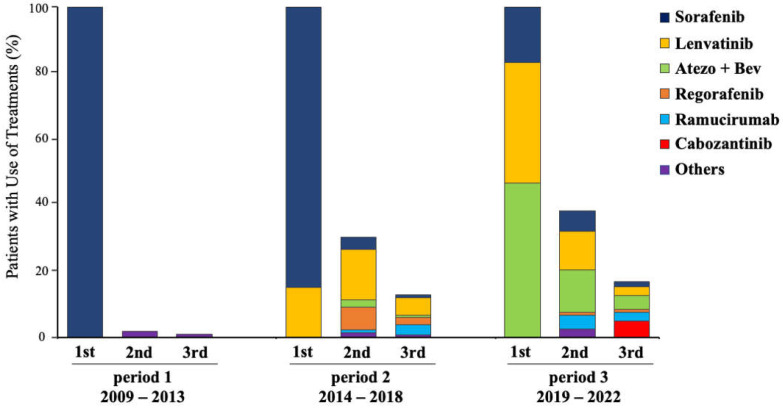
Systemic treatment use and proportion of patients with sequential therapy lines.

**Figure 3 cancers-15-05298-f003:**
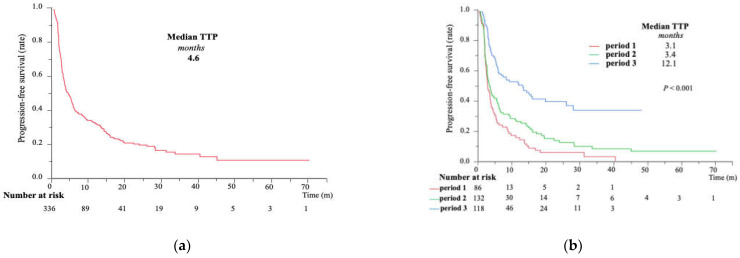
Time to progression of (**a**) the entire cohort and (**b**) the three period groups; period 1, 2009–2013; period 2, 2014–2018; period 3, 2019–2022, via modified RECIST. TTP, time to progression; mRECIST, modified Response Evaluation Criteria in Solid Tumors.

**Figure 4 cancers-15-05298-f004:**
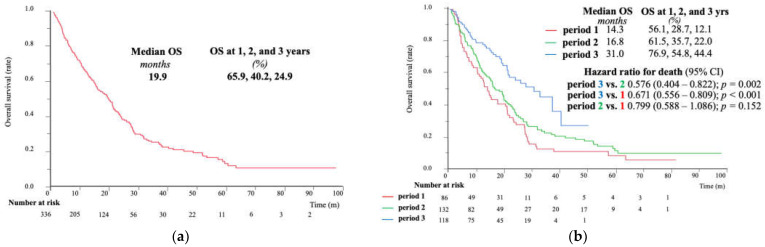
Overall survival of (**a**) the entire cohort and (**b**) the three period groups; period 1, 2009–2013; period 2, 2014–2018; period 3, 2019–2022. OS, overall survival.

**Figure 5 cancers-15-05298-f005:**
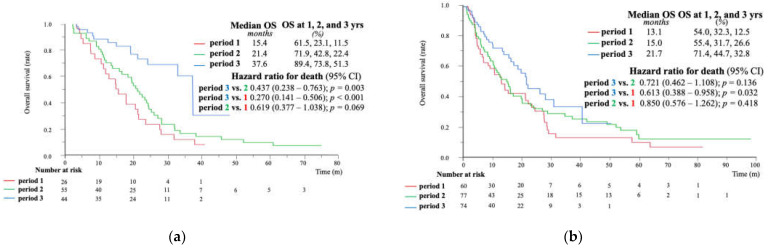
Overall survival in the three groups of (**a**) BCLC-B patients and (**b**) BCLC-C patients. BCLC, Barcelona Clinic Liver Cancer; OS, overall survival.

**Table 1 cancers-15-05298-t001:** Characteristics of the whole study population.

Variables	Category	All Patients
Number		336
Age (years)	median (range)	71 (19 to 89)
Sex, *n* (%)	Female	54 (16.1)
	Male	282 (83.9)
ECOG PS, *n* (%)	0	279 (83.0)
	1	49 (14.5)
	2	8 (2.5)
Body mass index (kg/m^2^)	median (range)	23.1 (11.2 to 32.8)
Etiology, *n* (%)	Viral	237 (71.0)
	Non-Viral	99 (29.0)
Child–Pugh class, *n* (%)	A	312 (92.9)
	B	24 (7.1)
ALBI score	median (range)	−2.54 (−3.61 to −0.91)
ALBI grade, *n* (%)	1 or 2a	234 (69.6)
	2b or 3	102 (30.4)
BCLC stage, *n* (%)	B	125 (37.2)
	C	211 (62.8)
Maximum intrahepatic tumor size, cm	median (range)	2.8 (0 to 25)
Intrahepatic tumor number, *n* (%)	≤4	156 (46.4)
	≥5	180 (53.6)
Macrovascular invasion, *n* (%)	Absent	251 (74.7)
	Present	85 (25.3)
Extrahepatic metastasis, *n* (%)	Absent	179 (53.3)
	Present	157 (46.7)
AFP (ng/mL)	median (range)	54 (0.9 to 1,057,992)
AFP ≥ 400 ng/mL, *n* (%)	Absent	224 (66.7)
	Present	112 (33.3)
The number of systemic therapy lines	1	248 (73.8)
	2	50 (14.9)
	3	24 (7.1)
	≥4	14 (4.2)

Abbreviations: ECOG PS, Eastern Cooperative Oncology Group Performance Status; ALBI, albumin–bilirubin; BCLC, Barcelona Clinic Liver Cancer stage; AFP, alpha-fetoprotein.

**Table 2 cancers-15-05298-t002:** Characteristics of the period 1, 2, and 3 groups.

Variables	Category	Period 1	Period 2	Period 3	*p*-Value
		(2009–2013)	(2014–2018)	(2019–2022)	
Number		86	132	118	
Age (years)	median (range)	67 (36 to 84)	71 (19 to 89)	71 (46 to891)	0.048
Sex, *n* (%)	Female	9 (10.5)	22 (16.7)	23 (19.5)	0.216
	Male	77 (89.5)	110 (83.3)	95 (80.5)	
ECOG PS, *n* (%)	0	74 (86.0)	108 (82.2)	97 (82.2)	0.743
	1	10 (11.6)	22 (16.7)	17 (14.4)	
	2	2 (2.4)	2 (1.5)	4 (3.4)	
Body mass index (kg/m^2^)	median (range)	22.2 (15.7 to 32.7)	22.9 (15.8 to 32.7)	24.1 (11.2 to 31.7)	0.005
Etiology, *n* (%)	Viral	72 (83.7)	93 (70.5)	72 (61.0)	0.002
	Non-Viral	14 (16.3)	39 (29.5)	46 (39.0)	
Child–Pugh class, *n* (%)	A	85 (98.8)	121 (91.7)	106 (89.8)	0.038
	B	1 (1.2)	11 (8.3)	12 (10.2)	
ALBI score	median (range)	−2.42 (−3.28 to −1.24)	−2.56 (−3.61 to −1.25)	−2.59 (−3.36 to −0.91)	0.137
ALBI grade, *n* (%)	1 or 2a	61 (70.9)	89 (67.4)	84 (71.2)	0.776
	2b or 3	25 (29.1)	43 (32.6)	34 (28.8)	
BCLC stage, *n* (%)	B	26 (30.2)	55 (41.7)	44 (37.3)	0.233
	C	60 (69.8)	77 (58.3)	74 (62.7)	
Maximum intrahepatic tumor size, (cm)	median (range)	2.8 (0 to 16.6)	2.6 (0 to 13)	3.5 (0 to 25)	0.034
Intrahepatic tumor number, *n* (%)	≤ 4	36 (41.9)	54 (40.9)	66 (55.9)	0.037
	≥ 5	50 (58.1)	78 (59.1)	52 (44.1)	
Macrovascular invasion, *n* (%)	Absent	58 (67.4)	105 (79.5)	88 (74.6)	0.141
	Present	28 (32.6)	26 (20.5)	30 (25.4)	
Extrahepatic metastasis, *n* (%)	Absent	40 (46.5)	69 (52.3)	70 (59.3)	0.186
	Present	46 (53.5)	63 (47.7)	48 (40.7)	
AFP (ng/mL)	median (range)	49 (2 to 1,057,992)	41.8 (1.3 to 189,050)	69.5 (0.9 to 129,880)	0.734
AFP ≥ 400 ng/mL, *n* (%)	Absent	57 (66.3)	91 (68.9)	76 (64.4)	0.584
	Present	29 (33.7)	41 (31.1)	42 (35.6)	
The number of systemic therapy lines	1	84 (97.6)	92 (69.7)	72 (61.0)	<0.001
	2	1 (1.2)	23 (17.4)	26 (22.0)	
	3	1 (1.2)	13 (9.9)	10 (8.5)	
	≥4	0	4 (3.0)	10 (8.5)	

Abbreviations: ECOG PS, Eastern Cooperative Oncology Group Performance Status; ALBI, albumin–bilirubin; BCLC, Barcelona Clinic Liver Cancer stage; AFP, alpha-fetoprotein.

**Table 3 cancers-15-05298-t003:** Therapeutic efficacy of the 1st-line therapy.

Therapeutic Efficacy	ALL	Period 1	Period 2	Period 3	*p*-Value
		(2009–2013)	(2014–2018)	(2019–2022)	
CR/PR/SD/PD/NE, *n*	9/60/73/136/58	1/10/12/51/12	4/16/23/61/28	4/34/38/24/18	<0.0001
ORR, *n* (%)	69 (20.5)	11 (14.9)	20 (19.2)	38 (38.0)	0.001
DCR, *n* (%)	142 (51.0)	23 (31.1)	43 (41.3)	76 (76.0)	<0.0001

Abbreviations: CR, complete response; PR, partial response; SD, stable disease; PD, progressive disease; NE, not evaluable; ORR, objective response rate; DCR, disease control rate.

**Table 4 cancers-15-05298-t004:** Prognostic factors associated with overall survival in the entire cohort.

Variables	Category	Univariate Analysis			Multivariate Analysis		
		HR	95% CI	*p*-Value	HR	95% CI	*p*-Value
Age (years)	<75	1		0.242	1		0.044
	≥75	1.182	0.893–1.566		1.355	1.008–1.820	
Sex	male	1		0.756	1		0.467
	female	0.942	0.647–1.372		0.865	0.586–1.278	
Etiology	viral	1		0.250	1		0.532
	non-viral	0.837	0.618–1.133		0.903	0.656–1.243	
Child–Pugh class	A	1		0.401			
	B	1.284	0.716–2.304				
ALBI grade	1 or 2a	1		<0.001	1		<0.001
	2b or 3	2.002	1.488–2.692		1.823	1.325–2.508	
BCLC grade	B	1		0.078	1		0.431
	C	1.283	0.972–1.694		0.801	0.461–1.392	
Maximum intrahepatic tumor size (cm)	≥5	1		0.212	1		0.099
	<5	0.880	0.721–1.075		1.033	0.994–1.075	
Intrahepatic tumor number	≤4	1		<0.002	1		<0.001
	≥5	1.548	1.176–2.039		1.859	1.327–2.604	
Macrovascular invasion	absent	1		0.002	1		0.079
	present	1.628	1.197–2.214		1.504	0.953–2.37	
Extrahepatic metastasis	absent	1		0.497	1		0.001
	present	1.098	0.838–1.438		2.297	1.378–3.828	
AFP (ng/mL)	<400	1		0.001	1		0.001
	≥400	1.615	1.225–2.129		1.622	1.216–2.164	
Number of systemic therapy lines	≥3	1		<0.0001	1		<0.001
	≤2	2.208	1438–3.389		2.776	1.751–4.402	
Therapeutic efficacy	OR (CR + PR)	1		<0.0001	1		<0.001
	non-OR	2.186	1.506–3.174		2.516	1.704–3.716	

Abbreviations: ECOG PS, Eastern Cooperative Oncology Group Performance Status; ALBI, albumin–bilirubin; BCLC, Barcelona Clinic Liver Cancer stage; AFP, alpha-fetoprotein; CR, complete response; PR, partial response; SD, stable disease; PD, progressive disease; NE, not evaluable; OR, objective response.

## Data Availability

The data that support the findings of this study are available from the corresponding author upon reasonable request.

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
