# Peer review of "Survival Improvements in Advanced Hepatocellular Carcinoma with Sequential Therapy by Era"

_cancers, 2023, doi:10.3390/cancers15215298_

Round 1
Reviewer 1 Report (Previous Reviewer 3)
Comments and Suggestions for Authors
I am writing to recommend the acceptance of the manuscript titled "Survival improvements in advanced hepatocellular carcinoma 2 with sequential therapy by era" submitted by Nakamura et al., for publication in [Journal Name]. Following a meticulous and comprehensive review, I am delighted to report that the manuscript meets all the journal's requirements and standards for publication without any need for changes or revisions.
The research presented in this article is of exceptional quality, demonstrating a clear and significant contribution to the field. The methodology is robust, the results are well-supported, and the writing is clear and engaging. This article aligns seamlessly with the objectives and scope of Cancers Journal.
I want to commend the author for their dedication to producing such a high-caliber piece of work. The paper is not only scientifically sound but also presents its findings in a manner that is accessible to a broad readership.
I believe that this article will make a valuable contribution to the scientific community and enhance the reputation of Cancers. I wholeheartedly recommend its immediate publication.
Author Response
Thank you so much for your kind and thoughtful comments, and taking time to review our article twice. We appreciate you so much.
Reviewer 2 Report (Previous Reviewer 1)
Comments and Suggestions for Authors
The authors investigated the survival improvement in advanced hepatocellular carcinoma (HCC) with sequential therapy by ara. Based on the background of the systemic therapy for HCC, they clearly showed that sequential treatment post-progression was connected to better survival. Moreover, they explained in more detail limitation of the study.
I have one minor concern.
Line 324:‘lenvatinib11’ is not right.
Author Response
Thank you so much for taking time to review our article twice, and pointing out our careless description.
As you pointed out, we deleted XX followed by 'lenvatinib', and corrected on page 11, line 324.
We appreciate you so much.
This manuscript is a resubmission of an earlier submission. The following is a list of the peer review reports and author responses from that submission.
Round 1
Reviewer 1 Report
Comments and Suggestions for Authors
The authors investigated the change during eleven years in survival for patients with unresectable HCC and effects of sequential systemic therapies on their outcomes. They concluded that sequential therapies are valuable for prolonging the survival. The retrospective study is well conducted. The data in presented clearly. However, there are some drawbacks in the manuscript.
1) Lines 3: I am somewhat skeptical for the title of the manuscript. Strictly speaking, the clinical data were analyzed during eleven years.
2) Lines 100-101: Please specify administered dose of regorafenib much as the other drugs.
3) Line101: Please change ‘6 mg/day’ into ‘60 mg/day’.
4) Lines 154: The proportion of patients in BCLC-A, who are not ideally candidate for systemic chemotherapy, should be stated.
5) Table 2: The ratios of intrahepatic tumor number and extrahepatic spread in period 1 tend to be larger than the other two periods. Moreover, the patient number in period 1 tends to be smaller than the other periods. The authors should clearly state the change in the times of treatment indication during eleven years in the discussion section.
6) Line 208: Please insert the phrase ‘in the entire cohort’.
7) Lines 293-294: Strictly speaking, the authors did not analyze only BCLC-B as mentioned above. Please elaborate.
Reviewer 2 Report
Comments and Suggestions for Authors
Thank you for the opportunity to review Survival improvements in advanced hepatocellular carcinoma with systemic therapy over the past decade by Nakamura et al. The study aimed to analyze the era-dependent trends in advanced HCC presentation and treatment as well as the role of sequential treatment in patient outcomes. Interpreting the results of the study is quite challenging. The cohort terminology is incongruent, as the target appeared to be advanced-stage therapy, but the patient population was quite diverse, from early-stage to advanced-stage. Further, era-based studies with multiple treatment lines are extremely complicated due to the overwhelming number of confounders and biases. Major concerns are outlined below:
Table 2 Between-period comparative statistics. A recommendation to improve the presentation of Table 2…if an overall group nonparametric significance test (Kruskal-Wallis) is P > 0.05 it may be easier to report the overall test P value. If the overall test is listed in the P value column, the intergroup differences could be designated with a superscript convention. The table is difficult to follow in the current format.
Era definitions. There should be a thorough explanation of the method that went into defining the eras.
Study title. I would highly recommend altering the manuscript title. The title reads like a review article. It is also missing the article highlight finding. I would highly recommend incorporating “sequential therapy” and specify “by decade / era” as opposed to “over the past decade”. At an absolute minimum, I would revise “over the past decade” to lose the review article tone in the title.
Available therapeutic lines over time. I worry there are some confounding interactions in the analysis that completely make sense but may not be appropriate to a forward-aiming prognostic analysis. For instance, the number of available treatment lines and regional availability of advanced oncology treatments have changed over time. That makes this a challenging variable to compare across large blocks of time because there are confounding factors built into it. Extending this, it’s more straightforward to compare the outcomes of sequential systemic therapy following loss of response or disease progression, but from the era-perspective, are we just comparing continuing treatment in the modern areas versus palliative care in the pre-modern era? I’m just not convinced there are some biases built in here that maybe just need some additional discussion.
Time of staging and treatment history in BCLC A-B. While there is no difference between BCLC A-B versus C across era, I have some concerns related to collating BCLC-A with BCLC-B and their overall prevalence in the study. This may perhaps relate to some deficiencies in the study methods. The study defines inclusion criteria as (a) imaging diagnosis and (b) receiving systemic therapy in the date range. Is the BCLC staging in the data assigned at (a) or (b)? My assumption is that the prevalence of BCLC A-B is related to staging as defined at diagnosis not at the initiation of systemic therapy. In the unresectable BCLC A-B, are these patients with a history of liver-directed therapy? If yes, this data should be disclosed and analyzed. This could have dramatic underlying effects on tumor biological aggressiveness that cannot be captured by, and more importantly are being obscured by, BCLC staging. If no, what was the reasons for proceeding directly to systemic therapy with such well-preserved liver function?
Lack of a viable second line in Era 1. I’m concerned as to how valid the multivariate Cox regression can be when sequential therapy is a variable that is essentially absent in Era 1.
Figure 2. The authors must devise an alternative labeling convention for Figure 2 or convert the figure to a table. It is challenging to identify the subgroups using only patterns and grayscale.
Table 3. I’m not sure we can discern much from this Table given the heterogeneity in first-line therapy and the expected overall improvement in response rate attributable to Lenvatinib and AtezoBev.
Figure 3. Is the sorafenib first-line ORR and duration of response stable across eras? If so, this could simply reflect the improved effectiveness of the first line agent diluted into an era designation.
Cox multivariate. I’m assuming keeping era in the multivariate was complicated due to limited observations for covariates across the eras. I feel like this study would have to focus solely on first line sorafenib to really be able to compare sequential therapy outcomes and it would have to control for first cycle response rate. As mentioned above, there are several confounders and biases built into the analysis.
Reviewer 3 Report
Comments and Suggestions for Authors
The current review manuscript by Nakamura et al. “Survival improvements in advanced hepatocellular carcinoma with systemic therapy over the past decade” showed immense potential and outstanding match for publishing in Cancers Journal. I found the research is up to the mark and recommend it for publication with very minor revisions. I feel that my below comments will further improve the manuscript and would be beneficial for the reader’s community of Cancers Journal.
My Detailed Review Comments are as follows:
1. Kindly provide the concise graphical abstract figure for a quick and proper understanding of the overall research manuscript work, which will gain more attention from the reader community of Cancers Journal.
2. Kindly check and correct the formatting and grammatical errors as per journal guidelines and standards.